# Photoredox-catalyzed diastereoselective dearomative prenylation and reverse-prenylation of electron-deficient indole derivatives

Xuexue Chang[1], Fangqing Zhang[1], Shibo Zhu[1], Zhuang Yang[2], Xiaoming Feng [1,3] ✉ & Yangbin Liu[1] ✉

Prenylated and reverse-prenylated indolines are privileged scaffolds in numerous naturally occurring indole alkaloids with a broad spectrum of important biological properties. Development of straightforward and stereo-selective methods to enable the synthesis of structurally diverse prenylated and reverse-prenylated indoline derivatives is highly desirable and challenging. In this context, the most direct approaches to achieve this goal generally rely on transition-metal-catalyzed dearomative allylic alkylation of electron-rich indoles. However, the electron-deficient indoles are much less explored, probably due to their diminished nucleophilicity. Herein, a photoredox-catalyzed tandem Giese radical addition/Ireland–Claisen rearrangement is disclosed. Diastereoselective dearomative prenylation and reverse-prenylation of electron-deficient indoles proceed smoothly under mild conditions. An array of tertiary α-silylamines as radical precursors is readily incorporated in 2,3-disubstituted indolines with high functional compatibility and excellent diastereoselectivity (>20:1 d.r.). The corresponding transformations of the secondary α-silylamines provide the biologically important lactam-fused indolines in one-pot synthesis. Subsequently, a plausible photoredox pathway is proposed based on control experiments. The preliminary bioactivity study reveals a potential anticancer property of these structurally appealing indolines.

In living organisms, dimethylallyl pyrophosphate (DMAPP) and iso-pentenyl pyrophosphate (IPP) as the precursors are usually utilized via enzyme catalysis to selectively introduce prenyl and reverse-prenyl motifs into various biological primary and secondary metabolites (Fig. 1a)[1–4]. As such, the post-translational proteins modified by a prenyl moiety are targeted to the correct membrane position and play a significant role in the biological signal transduction processes[5–7]. For small molecule metabolites, C3-prenylated and reverse-prenylated indoline scaffolds are frequently found in a variety of natural bioactive products[8–14], such as aszonalenin and flustramine, exhibiting a series of anticancer, antibacterial, and antifungal properties (Fig. 1b). Intrigued by their wide-ranging spectrum of biological activities, the construction of these dimethylallyl-related indoline derivatives has attracted intensive attention from synthetic chemists[15–29].

[1]Institute of Chemical Biology, Shenzhen Bay Laboratory, Shenzhen 518132, China. [2]State Key Laboratory of Biotherapy and Cancer Center, National Clinical Research Center for Geriatrics, West China Hospital of Sichuan University, Chengdu 610041, China. [3]Key Laboratory of Green Chemistry & Technology, Ministry of Education, College of Chemistry, Sichuan University, Chengdu 610064, China. ✉e-mail: xmfeng@scu.edu.cn; liuyb@szbl.ac.cn

**Fig. 1 | Synthesis of prenylated and reverse-prenylated indoline scaffolds.**
**a** Enzyme-catalyzed (reverse-)prenylation of complex molecules with isopentenyl pyrophosphate (IPP) or dimethylallyl pyrophosphate (DMAPP). **b** Representative naturally occurring prenylated and reverse-prenylated indoline products. **c** Previous work, transition metals-catalyzed allylic substituent reactions of electron-rich indoles. **d** This work, our designed strategy for the diastereoselective dearomative prenylation and reverse-prenylation of electron-deficient indoles via photocatalytic tandem Giese radical addition/Ireland–Claisen rearrangement. FG functional group, DG directing group, LG leaving group, PC photocatalyst, TMS trimethylsilyl.

Obviously, it is a formidable task bearing several key challenges to the design of synthetic strategies to access the prenylated and reverse-prenylated indolines: 1) unsymmetrical reaction sites of the dimethyl allyl groups often lead to a prominent regioselectivity problem with competitive prenylation (3,3-dimethylallyl) or reverse-prenylation (1,1-dimethylallyl); 2) the synthesis of C3-reverse-prenylated indolines is a more difficult process with relatively low reactivity, since two vicinal all-carbon quaternary centers present substantial steric hindrance; 3) to control the diastereoselectivity at the C3 and C2 sites of indoline also faces significant obstacles. In principle, catalytic dearomative prenylation of readily available indoles via allylic substituent reactions has been recognized as one of the most straightforward methods to access the dimethylallyl-related indolines in a single step (Fig. 1c)[30–35]. However, these corresponding transformations usually depend on the use of electron-rich indoles via an intramolecular dearomative process, and the precious metals (Ir and Pd) are often necessary. By contrast, the electron-deficient indoles are rarely employed as nucleophiles in the realm of allylic alkylation reactions, owing to the mismatched electrical properties. In addition, comparing with numerous advancements in the dearomatization of electron-rich indoles[36–40], the available dearomative reactions of electron-deficient indoles are much limited[41]. Therefore, it is distinctively challenging to achieve intermolecular dearomatization and prenylation of electron-deficient indoles simultaneously.

Recently, photoredox-enabled Giese-type radical addition[42–47] has been implemented to the dearomatization of electron-deficient indoles. In general, diverse radical precursors, such as tertiary amines, *N*-arylglycines, aliphatic carboxylic acids and aldehydes, are initialized by the excited photocatalysts, which undergo a radical nucleophilic addition to the electron poor C2 = C3 bond of indoles. The resulting dearomatized radicals are then reduced to give the corresponding anions, followed by a rapid protonation to deliver the hydrofunctionalized indoline derivatives (Fig. 1d, top)[48–56]. Inspired by our continuing interest in sigmatropic rearrangements[57–60], we envisage whether the incorporation of photoredox-enabled nucleophilic radical dearomatization of indoles and [3,3]-rearrangement (Ireland–Claisen type)[61–63] could be developed to prepare the prenylated and reverse-prenylated indolines (Fig. 1d, bottom). For instance, when employing 1-(3,3-dimethylallyl) indole-3-carboxylates as electrophilic substrates and α-silylamines as nucleophilic radical precursors, the generated in situ carbanions can be captured by TMS+, leading to the formation of highly reactive silylketene acetals. Subsequent [3,3]-rearrangement takes place under mild conditions and gives the C3-reverse-prenylated and C2-aminoalkylated indoline derivatives. Similarly, only by adjusting the 1-(1,1-dimethylallyl) substituents on electrophilic indoles, the complementary C3-prenylated indoline derivatives are obtained with ease and efficiency.

Herein, we demonstrate our efforts towards the visible-light-induced Giese radical dearomatization/Ireland–Claisen rearrangement with an organic photoredox catalyst, providing an alternative approach to accomplish the valuable prenylation and reverse-prenylation of electron-deficient indoles in good to excellent yields. An array of natural products and pharmaceuticals containing an aminoalkyl group are selectively incorporated in indoline scaffolds, displaying good tolerance of diverse functional groups. Distinct from a related work by Glorius using terminal acrylate esters as radical acceptors[64], here the prochiral C2-position of indoles will generate a new stereocenter after the radical addition, thus resulting in a challenging diastereoselective control for the subsequent [3,3]-rearrangement. Notably, after careful selection of *N*-directing groups and organic photocatalysts, the tandem Giese addition/rearrangement process was realized in an excellent diastereoselective manner, producing *trans*−2,3-difunctionalized indolines exclusively.

## Results

### Reaction optimization

To verify our strategy, the commercially available electron-deficient indole-3-carboxylic acid was converted to the corresponding 1-(3,3-dimethylallyl) ester, followed by a *N*-protection step to readily prepare the radical acceptor **1**. Considering the property of *N*-DG moiety can greatly affect the reactivity of indoles, we introduce diverse electron-withdrawing groups (such as Boc, Ac, Cbz, and Ts) on the nitrogen to decrease the electron density of indoles, thus making the indoles more reactive for the nucleophilic radical attack. Moreover, organic amines are prevalent motifs in a great variety of natural products and pharmaceuticals, which are used as the precursors for highly valuable α-aminoalkyl radicals under visible-light photoredox catalysis[65–79]. Consequently, we selected the *N*-Boc indole-ester **1a** and *N*-methyl-*N*-((trimethylsilyl)methyl)aniline **2a** as model substrates for the optimization of conditions. The mixture was irradiated by 1 W blue LEDs at room temperature and then heated at 60 °C to promote the thermal [3,3]-rearrangement process. After screening different photocatalysts, 4CzIPN was found to be the optimal catalyst, providing the dearomative reverse-prenylation product **3a** bearing vicinal all-carbon quaternary centers in 70% yield with moderate diastereoselectivity (3:1 d.r., Table 1, entry 4 vs entries 1–3). As expected, the valuable amine-moiety can be incorporated smoothly into indolines via Giese radical addition, however, a complicated diastereoselective control problem was accompanied for the sequential rearrangement. Further extensive optimization of solvents was investigated, and only slightly improved diastereomeric ratio (4:1 d.r.) was obtained in DMF without deterioration of the reactivity (78% yield, Table 1, entry 6). Then, we turned our attention to other *N*-protected indoles **1b**–**1d**. In general, the electrophilic indoles bearing *N*-electron withdrawing groups, such as Ac, Cbz, and Ts, were suitable substrates to afford the desired products in moderate to good yields. In contrast, no reaction occurred for *N*-H and *N*-Me indoles, indicating that the initial dearomatization process of the C2=C3 bond necessitated the activation of the *N*-EWGs. Encouragingly, when tosyl-substituted indole **1d** was employed, the reverse-prenylated indoline **3d** was furnished as a single diastereoisomer with >20:1 d.r. (Table 1, entry 12). Further optimization of the concentration of the reactants and the photocatalyst loading, we were delighted to isolate the indoline **3d** in 71% yield with exclusive diastereoselectivity (Table 1, entry 13). Of note, small amounts of the Giese-type side products (hydro-aminoalkylation of indoles) were identified by ¹H NMR analysis, probably due to the inevitable water in the reaction system. Additionally, confirmed by the control experiments, no reaction was observed in the absence of photocatalyst or light, indicating that a visible light mediated pathway was definitely involved (Table 1, entries 14 and 15).

### Access to reverse-prenylated indolines

With the optimized conditions in hand, we began to explore the generality of this dearomative reverse-prenylation with respect to electrophilic 1-(3,3-dimethylallyl) indole-3-carbonate **1d** and a range of functionalized tertiary α-silylamines **2** (Fig. 2). First, structurally diversified *N*,*N*-dimethyl(alkyl) anilines and *N*,*N*-diaryl amines were well tolerated to give the corresponding reverse-prenylated indolines (**3d**–**3n**) in good yields with excellent diastereoselectivities (55−90% yields, >20:1 d.r.). Electron-withdrawing and electron-donating substituents on varied positions of the aryl moiety were found to be compatible, including halogens, cyano, and methyl. The structure of reverse-prenylated indoline was determined by X-ray crystallographic analysis (**3h**, CCDC 2225579), demonstrating excellent *trans*-selectivity for the aminoalkyl and reverse-prenyl. Additionally, aniline derivatives bearing allyl, benzyl, and isopropyl groups exhibited exclusive terminal-methyl regioselectivity, as observed in **3j**–**3l**. Carbazole as an important core structure found in numerous natural compounds was also introduced into this catalytic system and showed modest reactivity (**3o**, 50% yield). We also investigated the acyclic aliphatic amines: *N*-methylbenzyl amine and exchange of the benzyl moiety with phenethyl smoothly afforded the desired 2,3-disubstituted indolines **3p** and **3q**. Subsequently, the cyclic aliphatic amines as the most frequent *N*-heterocyclic functions employed in drugs were examined. As such, substituted piperidines with varying functionalities (**3r** and **3s**) demonstrated good yields in 76% and 69%, respectively. Other relevant cyclic amines bearing additional heteroatoms and functionalities, including morpholine, piperazine, 1,4-diazepane, and L-proline, also underwent dearomatization smoothly and furnished the rearranged products **3t**–**3w** with analogous efficiencies. Substituted indoles possessing various functional groups on varied positions were also suitable substrates to deliver the reverse-prenylated indolines in good yields with excellent diastereoselectivities (46−73% yields, >20:1 d.r., **3x**−**3ae**). Given the tertiary amines as a common motif in pharmaceuticals, we were triggered to explore the capacity of this dearomative reverse-prenylation manifold on a range of pharmaceutically relevant compounds. As such, structurally complex drugs, including antibacterial infection norfloxacin (**3da**), acetylcholine receptor agonist cytisine (**3db**), calcium-channel blocker nortriptyline (**3dc**), and serotonin reuptake inhibitor duloxetine (**3dd**), were readily and selectively cross-coupled with indolines by the standard protocol. Alternatively, the significant amino-fragments found in other complex drugs such as ramipril (**3de**), flunarizine (**3df**), clopidogrel (**3dg**), and bepotastine (**3dh**), also afforded the corresponding indoline-adducts in good yields. Collectively, our catalytic platform provides a streamlined synthesis of structurally diverse reverse-prenylated indoline derivatives bearing a series of high-value-added amino-functionalities.

### Access to lactam-fused indolines

We next investigated secondary α-silylamines bearing a free NH group as radical donors under optimal conditions (Fig. 3). Surprisingly, the unexpected lactam-fused indolines became the major products with excellent diastereoselectivities. We speculate that the acidic TMS⁺ in the reaction mixture may activate the indoline-3-carboxylic acid and facilitate the intramolecular amidation. Given the distinct advantage of lactam scaffolds in pharmaceuticals[80,81], we examined the generality of this one-pot synthesis of lactam-fused indolines. Good reactivity for a variety of secondary α-silylamines with diverse functional groups was displayed, including halogens, *tert*-butyl, ester, trifluoromethyl, and methoxyl (**5a**−**5h**). Also, the three-dimensional structure of compound **5f** was confirmed by X-ray diffraction analysis (CCDC 2225587). Furthermore, amino acids derivatives, glycine and L-methionine, were also applied to deliver the corresponding lactam-fused indolines **5i** and **5j** in good yields. Product **5j** was provided as two diastereoisomers due to the presence of an extra stereocenter.

**Table 1 | Reaction optimization for dearomative reverse-prenylation of indoles**

| entry[a] | substrate | PC | solvent | yield (%)[b] | d.r.[c] |
|---|---|---|---|---|---|
| 1 | 1a | Ru(bpy)$_3$(PF$_6$)$_2$ | CH$_3$CN | 60 | 2.8:1 |
| 2 | 1a | Ir(bpy)$_2$(dtbbpy)PF$_6$ | CH$_3$CN | 76 | 2.5:1 |
| 3 | 1a | Mes-Acr⁺PF$_6^-$ | CH$_3$CN | trace | -- |
| 4 | 1a | 4CzIPN | CH$_3$CN | 70 | 3:1 |
| 5[d] | 1a | 4CzIPN | THF | 75 | 3:1 |
| 6 | 1a | 4CzIPN | DMF | 78 | 4:1 |
| 7 | 1a | 4CzIPN | DMSO | 80 | 3:1 |
| 8 | 1a | 4CzIPN | EtOAc | 24 | 6:1 |
| 9 | 1a | 4CzIPN | toluene | trace | -- |
| 10 | 1b | 4CzIPN | DMF | 80 | 2.4:1 |
| 11 | 1c | 4CzIPN | DMF | 30 | 2:1 |
| 12 | 1d | 4CzIPN | DMF | 65 | >20:1 |
| 13[e] | 1d | 4CzIPN | DMF | 71[f] | >20:1 |
| 14[g] | 1d | -- | DMF | 0 | N.D. |
| 15[h] | 1d | 4CzIPN | DMF | 0 | N.D. |

[a] Reaction conditions: indoles 1 (0.1 mmol), α-silylamine 2a (0.12 mmol), and photocatalyst (2.5 mol%) were performed in the indicated solvent (1.0 mL) at room temperature under the irradiation by 1 W blue LEDs for 2 h in argon. Then, the reaction vial was warmed at 60 °C for additional 3 h without blue LEDs.
[b] Yield determined by $^1$H NMR with pyridine as the internal standard.
[c] Diastereomeric ratio determined by $^1$H NMR analysis of the crude reaction mixture.
[d] Irradiation time was extended to 48 h.
[e] Indoles 1d (0.3 mmol), α-silylamine 2a (0.36 mmol), 4CzIPN (1.3 mol%), DMF (1.0 mL).
[f] Isolated yield.
[g] No photocatalyst.
[h] Under dark. PC, photocatalyst; N.D., not determined.

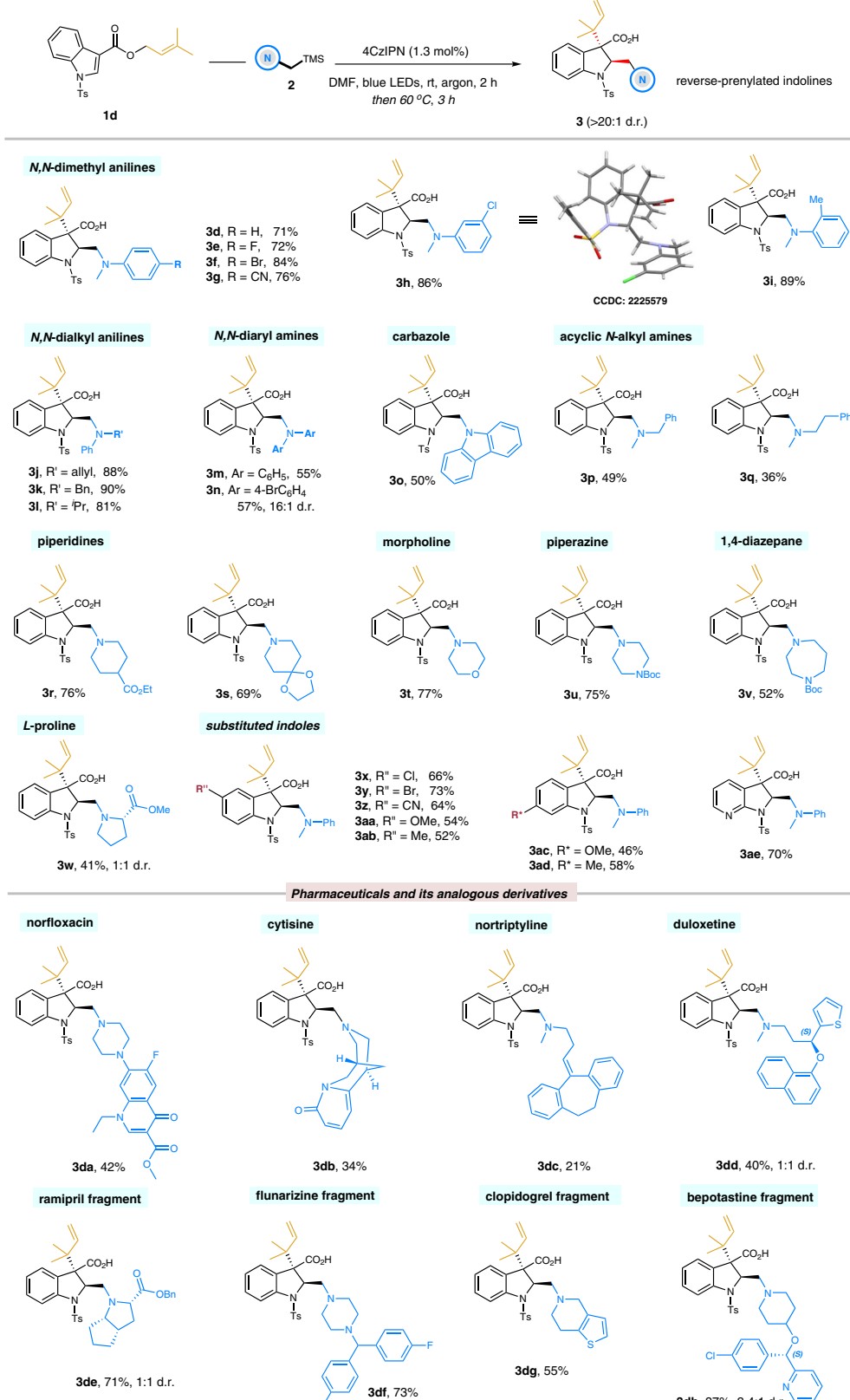

**Fig. 2 | Substrate scope of reverse-prenylated indolines.** Reaction conditions: solution of **1d** (0.3 mmol), **2** (0.36 mmol), and 4CzIPN (1.3 mol%) in DMF (1.0 mL) was irradiated by 1 W blue LEDs for 2 h in argon. Then the reaction vial was warmed at 60 °C for additional 3 h without blue LEDs. Isolated yields are shown. d.r. values were based on $^1$H NMR analysis.

**Fig. 3 | Substrate scope of the lactam-fused indolines from various secondary α-silylamines.** Reaction conditions: solution of **1d** (0.3 mmol), **4** (0.36 mmol), and 4CzIPN (1.3 mol%) in DMF (1.0 mL) was irradiated by 1 W blue LEDs for 2 h in argon. Then the reaction vial was warmed at 60 °C for additional 3 h without blue LEDs. Isolated yields are shown. d.r. values were based on [1]H NMR analysis.

## Access to prenylated indolines

Next, we wondered if the complementary regioisomers, prenylated indolines, could be prepared via this uniform dearomatization-rearrangement sequence (Fig. 4). Delightfully, by simply changing the ester group at the C3 position of indoles into 1-(1,1-dimethylallyl) substituent (**6**), the expected transformations proceeded smoothly and afforded the prenylated indolines with good yields and excellent diastereoselectivities (**7a–7j**). A wide of tertiary α-silylamines bearing various functionalities and diverse structures, such as N,N-dialkyl anilines, carbazole, morpholine, piperidine and piperazines, were well tolerated and their corresponding prenylated indolines **7a–7h** were obtained in good to excellent yields (66–83%) with exclusive selectivity (> 20:1 d.r.). Similarly, when using secondary α-silylamines as the radical precursors, the lactam-fused indoline analogues **7i** and **7j** containing a C3-prenyl substituent were achieved with comparable results.

## Access to *trans*−2,3-disubstituted indolines

In addition, other complex terpene-derived indole-3-carboxylates were found to be suitable substrates for this reaction (Fig. 5). As such, the photoredox-enabled dearomative reverse-geranylation/far-nesylation proceeded smoothly under the standard conditions, giving the corresponding products **9a** and **9b** in moderate yields (59–66%), but with low diastereomeric ratios (2:1 and 1.3:1 d.r.). It is probably attributed to the difficulty in discriminating between the chairlike and boatlike transition states in the Ireland−Claisen rearrangement process. Also, the readily available (Z)-pent-2-en-1-ol derivative was investigated to afford the 2,3-disubstituted indoline **9c** with a similar level of diastereoselectivity (89% yield, 1.7:1 d.r.). Furthermore, different substituents in the allyl alcohols were also tested, such as hydrogen, methyl and chloro-groups. Gratifyingly, the corresponding products **9d–9f** can be obtained in good yields with exclusive diastereoselectivities. It was worthy of noting that the propargylic alcohol-derived indole was well tolerated, delivering the desired indoline **9g** with an allenyl-substituted quaternary center in good outcomes (58% yield, >20:1 d.r.).

## Synthetical application

To further demonstrate the synthetic potential of this mild dearomatization-rearrangement reaction of electrophilic indoles, we next explored divergent transformations of taking the reverse-

prenylated products as an example (Fig. 6). Benefiting from the presence of diverse functional groups, such as terminal olefin, carboxylic acid, and amino, a variety of valuable synthetic building blocks were readily and selectively accessed after simple operations. Hydrogenation of the terminal olefin in reverse-prenyl moiety with palladium under a hydrogen atmosphere furnished **10** in 90% yield. Spiro-lactone **11** containing a primary alkyl iodide was directly prepared in 63% yield with excellent diastereoselectivity by an iodine-mediated halolactonization. Additionally, after an almost quantitative methyl-esterification, the olefin functionality was smoothly converted into primary alcohol **12** and organoboron **13** by a sequential hydroboration-oxidation (9-BBN and then $H_2O_2$), and an iridium-catalyzed hydroboration with pinacolborane, respectively. The synthetic utility of these indoline derivatives has also been shown to perform late-stage modifications of natural products to give their indoline-based analogous (for instance, estrone, **14**). The carboxylic acid group was also amenable to deliver the corresponding alcohol **15** under common conditions (reduction by LAH). Interestingly, when trying to prepare the acyl chloride from classical oxalyl chloride and DMF, an unexpected intramolecular demethylation-amidation reaction of the tertiary amine happened, resulting in the formation of lactam-fused indoline **5a** in 80% yield. Finally, deprotection of the N-tosyl indoline was accomplished by the single electron transfer reagent (sodium napthalenide), delivering the N-H free indoline derivative **16** in 64% yield.

## Mechanistic studies

To gain more insight into the reaction, preliminary mechanistic studies were performed. The Stern−Volmer fluorescence quenching experiments were conducted for each component **1d** and **2a** (Fig. 7a). It was found that the excited photocatalyst 4CzIPN* was quenched significantly by the α-silylaniline **2a** rather than the indole derivative **1d**, which suggested the oxidative single-electron transfer of α-silylanilines probably triggered the photoreaction. Interestingly, the fluorescence intensity was obviously enhanced in the presence of **1d**, implying a possible interaction between the photocatalyst and the indole substrate[82]. When 2,2,6,6-tetramethylpiperidinyloxy (TEMPO) and allylic sulfoxide were used as radical-trapping reagents, the desired dearomatizaiton/rearrangement product **3** was dramatically suppressed (Fig. 7b). Moreover, the α-allylated N,N-dimethyl aniline (**2a-allyl**) was confirmed by ESI-HRMS (m/z calcd for $C_{11}H_{15}N$ [M + H]$^+$:

**Fig. 4 | Substrate scope of prenylated indolines.** Reaction conditions: solution of **6** (0.3 mmol), **2** (0.36 mmol), and 4CzIPN (1.3 mol%) in DMF (1.0 mL) was irradiated by 1 W blue LEDs for 2 h in argon. Then the reaction vial was warmed at 60 °C for additional 3 h without blue LEDs. Isolated yields are shown.

162.1277; found: 162.1278), indicating that the aminoalkyl radical might be involved in this transformation (Fig. 7b). Next, *N,N*-dimethyl aniline **2′** was employed as the nucleophilic radical source and exclusively gave the hydro-aminoalkylated product **3′** (Giese-type) in 36% yield, which might be attributed to the rapid protonation of the C3-benzylic anion (Fig. 7c, first). Similarly, once external H$_2$O as a proton source was introduced into the standard reaction, the Giese-type side product **3′** became dominated and the [3,3]-rearrangement process was completely interrupted, further evidencing the generation of a carbanion in this reaction (Fig. 7c, second). To probe the reactive silylketene acetal intermediate, we modified the reaction conditions by switching DMF to CH$_3$CN and reducing room temperature to 0 °C. In that case, the spontaneously thermal [3,3]-rearrangement became negligible, and the ESI-HRMS signal of **3′** + TMS (m/z calcd for C$_{32}$H$_{40}$N$_2$O$_4$SSi [M + H]$^+$: 577.2551; found: 577.2553) was clearly observed (Fig. 7c, third). Accordingly, on the basis of the above control experiments along with previous works[48–52], a plausible reaction mechanism was proposed (Fig. 7d): firstly, the excited 4CzIPN* ($E_{1/2}$(PC*/PC$^-$) = +1.35 V vs. SCE in MeCN)[83] underwent a SET oxidation of the α-silylaniline **2** ($E_{1/2}$(**2a**$^{+}$/**2a**) = approx. +0.7 V vs. SCE in MeCN)[84], followed by fragmentation to afford α-aminoalkyl radical **I** and TMS$^+$. Next, the nucleophilic radical **I** attacked *N*-Ts indole derivative **1** in a similar Giese reaction pathway to generate C3-benzylic radical **II**, which was then reduced by the 4CzIPN$^-$ to give a C3-benzylic anion **III**. Subsequent enolization of the anion **III** provided in situ the silylketene acetal intermediate **IV**. Finally, a spontaneous Ireland–Claisen rearrangement of **IV** furnished the dearomative reverse-prenylation of indoles, to

deliver the product **3** in good yield with exclusive *trans*-selectivity of the newly formed C-C bonds (red color highlighted in **3**). We speculated that the *cis*-C3-allyl migration might be blocked by the C2-aminoalkyl substituent in transition state **IV-2**.

### Preliminary study on the anticancer activity
Considering prenylated/reverse-prenylated indolines as an important motif in bioactive natural products, we initially investigated the anticancer activity of several selected indolines by detecting the in vitro cytotoxicity against the human leukemia cell line MV4-11 (Fig. 8). It was found that most candidates had a preliminary inhibition effect at 10 μM. Notably, compounds **3k** and **3df** displayed a potential anticancer activity with IC$_{50}$ values of 16.8 μM and 6.3 μM, respectively. We believe that further study of other indoline derivatives and biological properties is promising to discover more potential applications in medicinal chemistry.

### Discussion
In conclusion, we have developed an intermolecular dearomative prenylation and reverse-prenylation of indoles via a tandem Giese radical addition/Ireland–Claisen rearrangement. Distinct from conventional allylic substitution approaches mostly relying on electron-rich indoles, this photoredox-enabled protocol bearing transition-metal-free provides an efficient method to achieve (reverse-)prenylation of electron-deficient indoles. Moreover, after careful selection of organic photocatalysts and *N*-protection moieties, the 2,3-difunctionalized indoline derivatives were produced with exclusive

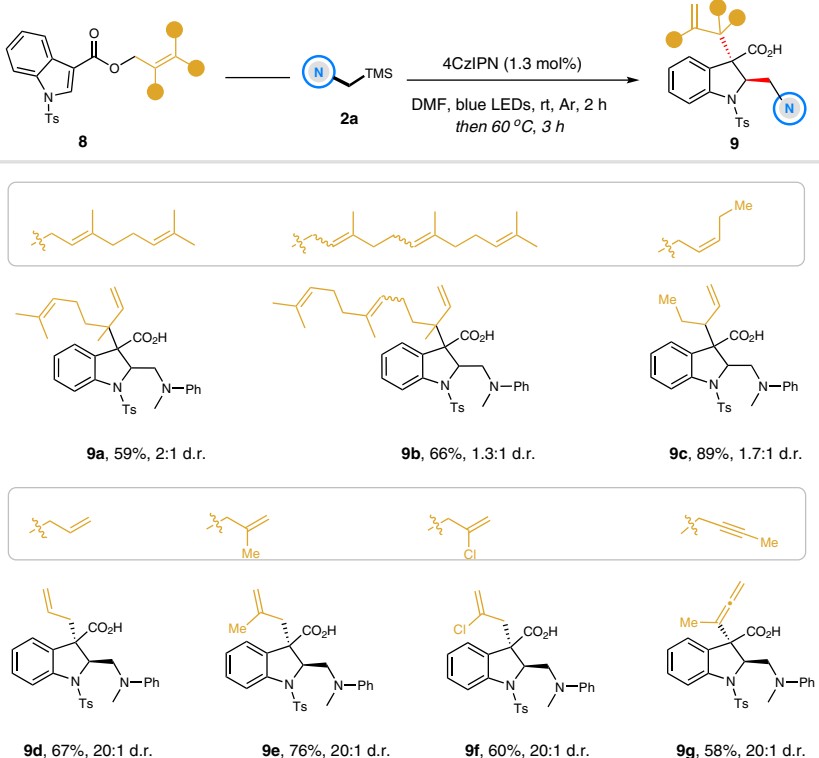

**Fig. 5 | Preparation of other relevant trans-2,3-disubstituted indolines.** Reaction conditions: solution of **8** (0.3 mmol), **2a** (0.36 mmol), and 4CzIPN (1.3 mol%) in DMF (1.0 mL) was irradiated by 1 W blue LEDs for 2 h in argon. Then the reaction vial was warmed at 60 °C for additional 3 h without blue LEDs. Isolated yields are shown. d.r. values were based on ¹H NMR analysis.

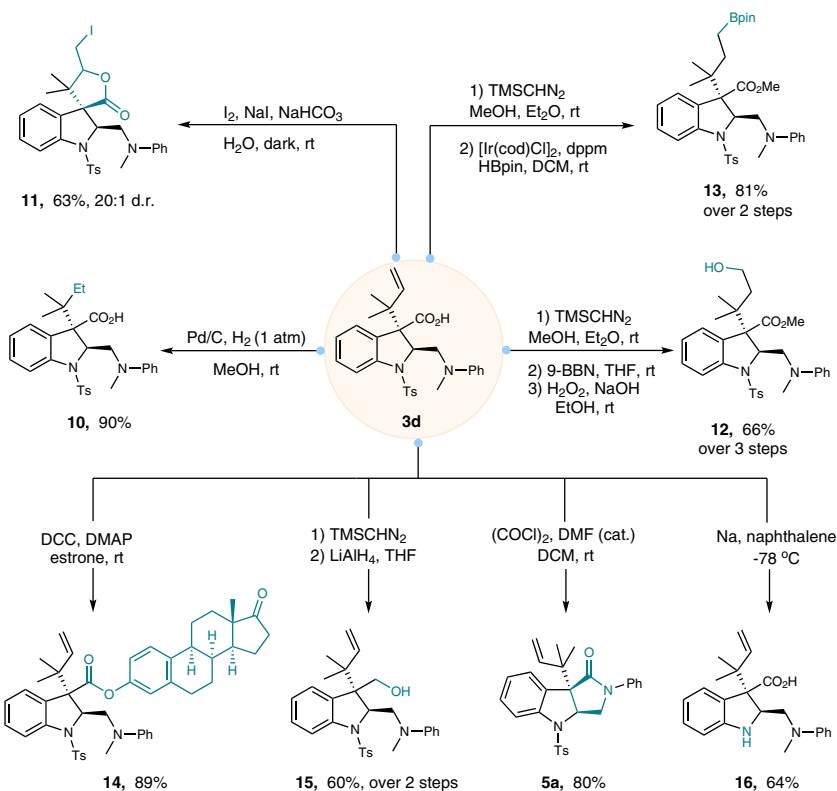

**Fig. 6 | Synthetic transformations of the reverse-prenylated indolines.** Conversion of the enantioenriched product **3d** to diverse chiral building blocks (for details, see Supplementary information).

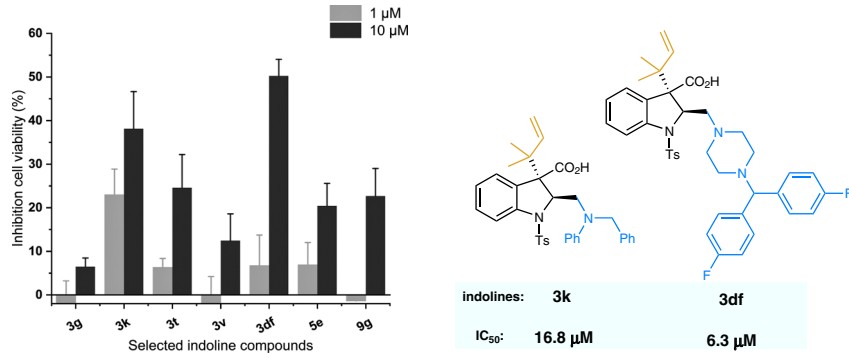

**Fig. 7 | Mechanistic investigation. a** Stern–Volmer fluorescence quenching experiments. **b** Radical trapping experiments. **c** Intermediate confirmation experiments. **d** Proposed reaction mechanism.

**Fig. 8 | Biological activity study.** Inhibition of the human leukemia cell line viability (MV4-11) induced by some selected indoline compounds. IC$_{50}$ values for **3k** and **3df** in inhibiting MV4-11. Experiments were performed in duplicate ($n = 3$). Standard deviation (SD) values are shown as the error bars.

diastereoselectivities (>20:1 d.r.). An array of structurally diverse amines including complex modified natural products and pharmaceuticals were employed as radical precursors, and were readily incorporated in indolines with high functional compatibility and isolated yields. Notably, by simple adjustment of 1-(1,1-dimethylallyl) or 1-(3,3-dimethylallyl) substituent in indole-3-carboxylates, both prenylated and reverse-prenylated indolines were selectively produced, respectively, without the disturbing regioselectivity issue. In addition, the current systems were found to work well with the secondary amines, efficiently affording the biologically important lactam-fused indolines in one-pot synthesis. The synthetic potential was further highlighted via diversification of reverse-prenylated products. Mechanistic studies revealed a possible photoredox-SET process, followed by the formation of silylketene acetals and subsequent [3,3]-rearrangement. Finally, the anticancer activity of these privileged indoline products was preliminarily explored, indicating potential application prospects in biological activity.

## Methods

### Materials

Unless otherwise specified, all chemicals were purchased from Leyan.com and Bide Pharmatech. All solvents were purified and dried according to standard methods before use.

### General procedure for the photoredox-catalyzed dearomative prenylation and reverse-prenylation of electron-deficient indoles

In the glovebox, to a flame-dried 8 mL reaction vial equipped with a stir bar were added dimethylallyl indole-3-carboxylate (0.3 mmol, 1.0 equiv.) and 4-CzIPN (3.0 mg, 0.0039 mmol, 0.013 equiv.) in dry DMF (0.5 mL). Then the solution of α-silylamine (0.36 mmol, 1.2 equiv.) in dry DMF (0.5 mL) was added. The vial was sealed and transferred out of the glove box. It was irradiated with a 1 W blue LED lamp (SYNLED) for 2 h at room temperature. Afterwards, the reaction mixture was allowed to heat at 60 °C for 3 h without light. When the reaction was completed (monitored by TLC), the crude mixture was quenched by water and extracted with ethyl acetate (10 mL × 2). The combined organic layers were washed with water (10 mL × 2) and brine (10 mL), dried over anhydrous $Na_2SO_4$, filtered, and concentrated by rotary evaporation. Then the residue was purified by silica gel flash chromatography to give the corresponding product.

## Data availability

Crystallographic data for the structures reported in this Article have been deposited at the Cambridge Crystallographic Data Centre, under deposition numbers 2225579 (**3h**) and 2225587 (**5f**). Copies of the data can be obtained free of charge via https://www.ccdc.cam.ac.uk/structures/. All other data supporting the findings of this study are available from its Supplementary information or the corresponding author upon request.

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

## Acknowledgements

We appreciate the National Natural Science Foundation of China (22001177), Shenzhen Bay Laboratory (S201100003 and S211101001-1), Shenzhen Bay Qihang Fellow (QH23001), Guangdong Pearl River Talent Program (2021QN020268) for generous financial support. We also thank Dr. Zhenda Tan and Dr. Hongkai Wang for helpful discussions.

## Author contributions

X.X.C. performed the experiments and prepared the Supplementary information. F.Q.Z. investigated several substrates and repeated some data. S.B.Z. developed the initial reaction conditions. Z.Y. performed the biological experiments. X.M.F. supported the project. Y.B.L. designed and supervised the project, and wrote the manuscript and Supplementary information. All the authors contributed to the manuscript revisions.

## Competing interests

The authors declare no competing interests.
