## [Peer Review File · Nature Communications]

REVIEWER COMMENTS

Reviewer #1 (Remarks to the Author):

Nature employs dimethylallyl pyrophosphate (DMAPP) and isopentenyl pyrophosphate (IPP)[1] as starting materials for the biosynthesis of numerous prenylated and reverse-prenylated indole alkaloids. In this manuscript, Liu and co-workers developed a photoredox-catalyzed intermolecular dearomative indoles via a tandem Giese radical addition/Ireland-Claisen rearrangement. Prenylated and reverse-prenylated groups could be incorporated onto electron-deficient indoles in high region- and diastereoselectivities. A lot of structurally diverse amines including complex modified natural products and pharmaceuticals were employed as radical precursors to demonstrate the functional compatibility of this protocol. Preliminary mechanistic studies have been performed to support the proposed mechanism. Furthermore, some synthetic transformations and biological activities of the obtained product have been shown to highlight its potential applications. Overall, this manuscript is recommended to be accepted for publication in Nature Communications after some revisions.

Questions and Suggestions:

- 1) The authors showed good substrate scope for amine precursors. However, only two examples have been shown for substituted indoles (3x, 3y). It is better to give more examples for substituted indoles.
- 2) Figure 6: The yield for product 12 should refer to the yield of the three steps.
- 3) Figure 7a should be shown in higher resolution.
- 4) For readers, it is better to show the structure of 1d, 2a, 3d once in the Figure 7b. And the compound labelings for 1d, 2a, 3d should be corrected as 1, 2, 3.
- 5) SI: There are many mistakes with the coupling constant of ¹H NMR, for example, product 1y 7.83 (t, J = 8.8 Hz, 3H), 3d 4.82 (dd, J = 21.4, 14.2 Hz, 2H), 3g 4.97 (dd, J = 32.4, 14.2 Hz, 2H), 3h 4.96 (dd, J = 21.8, 1.2 Hz, 2H), 3u 3.56 (d, J = 26.7 Hz, 4H), 3dc 1.01 (d, J = 5.2 Hz, 3H), 3dd 0.98 (d, J = 9.6 Hz, 3H), -0.02 (d, J = 16.5 Hz, 3H), 7a 7.67 (dd, J = 10.8, 8.3 Hz, 3H), 7.18 (d, J = 7.9 Hz, 3H), 6.88 (dd, J = 17.5, 7.8 Hz, 3H), 7h 7.33 (dd, J = 13.8, 8.3 Hz, 4H), 7j 4.35 (d, J = 17.6 Hz, 1H). Please check carefully through SI.
- 6) SI: Some coupling constants between ¹⁹F-¹³C are missing in ¹³C NMR data, for example, compounds 5b, 5g.
- 7) SI: There are some mistakes for MS in SI, for example, "m/z [M+H]⁺ calcd for C₃₁H₄₁N₃O₆S: 584.2789..." in page 29. Please check carefully.
- 8) Related works on the catalytic prenylation and reverse-prenylation of indoles, such as *Angew. Chem. Int. Ed.* 2019, 58, 5438, are suggested to cite.

Reviewer #2 (Remarks to the Author):

The work "Photoredox-Catalysed Diastereoselective Dearomative Prenylation and Reverse-Prenylation of Electron-Deficient Indole Derivatives" by Feng, Liu and co-workers is a comprehensive study on the derivatization of electron-poor indoles. The strategy allows to obtain dearomatization and functionalization of two sites of the indole with high diastereoselectivity in a one-pot reaction.

The protecting group role was thoroughly investigated, and a tosyl substitution identified as the best to achieve high diastereoselectivity.

The mechanistic studies confirm a photoredox-catalyzed mechanism for the first reaction pathway, then followed by an Ireland-Claisen rearrangement. The radical attack on electron-poor indoles has received notable attention in the field of photoredox catalysis. Despite being overall a non-novel reaction pathway, the end product renders the work ingenious and of remarkable utility when considering possible further modifications, as demonstrated by the authors in Figure 6.

On contrary, the Ireland-Claisen rearrangement does not work with tertiary amine as a radical substrate. It would be interesting to see some other radical precursors viz. acids, boronic ester, or trifluoro borates as controls! Further, since the chosen radical precursors of choice were silanes, what about benzyl or alkyl ones?

The scope is also well-studied and broad. As indicated in the text, electron-poor indoles are required for the transformation. When considering substitution patterns on indoles, the authors only report two halogen substituted structures (3x and 3y). Are those the only tolerated substitutions? How do more electron-rich groups in those positions behave? Some additional references on radical indole difunctionalizations are missing (e.g. Nat Commun. 2020, 11, 3263, Org. Lett. 2022, 24, 9386...).

The SI is complete and reports full synthetic procedures and characterization of starting materials and final products.

Reviewer #3 (Remarks to the Author):

The authors reported an interesting photoredox-catalysed tandem Giese radical addition/Ireland-Claisen rearrangement, providing an efficient method to achieve (reverse-)prenylation of electron-

deficient indoles. The methods reported look to be effective to achieve the scaffolds described. The paper is certainly detailed, and a lot of work has been done. The paper is well written and presented, and the quality of the work looks to be high. My biggest concern is the novelty since the idea of dearomative prenylation of indole derivatives has been reported several times. And the advantages of this method are not obvious compared with previously reported methods. Therefore, the manuscript is not recommended to be published in Nature Communications. Detailed comments are as follows:

1. You and co-authors have realized the enantioselective dearomative prenylation of indole derivatives (nature catalysis, 2018, 1, 601), while this work only achieved the diastereoselective dearomative prenylation of indole derivatives.
2. Although the method uses photocatalysis, it is actually more cumbersome than the reported methods. The method is a two-step continuous reaction, but the second step needs to change the reaction conditions, and it need esterification and N-protection steps to readily prepare the radical acceptor.
3. The methods of Carreira, You and Stark have realized the construction of complex fused-heterocycle skeletons and even natural products, while this method is only suitable for the construction of indoline derivatives with simple structure.
4. The authors emphasizes "Conventional approaches depend on the use of electron-rich indoles, this method achieved (reverse-)prenylation of electron-deficient indoles", in fact, the reaction process of this method requires the participation of carbonyl group on indoles, its reaction mechanism depends on the carbonyl group. This is also a limitation of the method because it is not applicable to the substrate without carbonyl group. Other methods have different reaction mechanisms, which do not mean that the dearomatization of electron-deficient indoles is difficult to achieve.
5. The antitumor activity of the synthesized indolines are too weak ($IC_{50} > 5 \mu M$) compared with reported kinase inhibitors ($IC_{50} < 50 nM$) for the human leukemia. The description of "compounds 3k and 3df displayed superior anticancer activity" is not appropriate.

In view of the above, this paper is not suitable for publication in Nature Communications. It is good work however, and publication elsewhere would be appropriate with minor changes.

Reply to the comments by reviewer 1

1. The authors showed good substrate scope for amine precursors. However, only two examples have been shown for substituted indoles (**3x**, **3y**). It is better to give more examples for substituted indoles.

Response: Thanks for your positive comments and kind suggestions. To further demonstrate the substrate scope, more examples for substituted indoles were investigated, including electron-rich and electron poor groups (5-Cl, 5-Br, 5-CN, 5-OMe, 5-Me, 6-OMe, 6-Me and 7-azaindole). The corresponding reverse-prenylated indolines were obtained in good yields with exclusive diastereoselectivity (46–73% yields, >20:1 d.r., **3x–3ae**).

substituted indoles

2. Figure 6: The yield for product **12** should refer to the yield of the three steps.

Response: We have revised the yield for product **12** for over 3 steps.

3. Figure 7a should be shown in higher resolution.

Response: Thanks for your kind suggestions. We have tried to improve the resolution in Figure 7a.

4. For readers, it is better to show the structure of **1d**, **2a**, **3d** once in the Figure 7b. And the compound labelings for **1d**, **2a**, **3d** should be corrected as **1**, **2**, **3**.

Response: Thanks for your kind suggestions. We have revised the labeling in Figure 7b.

5. SI: There are many mistakes with the coupling constant of ¹H NMR.

Response: Thanks for your reminder. We have checked all the coupling constants of ¹H NMR.

6. SI: Some coupling constants between ¹⁹F-¹³C are missing in ¹³C NMR data.

Response: Thanks for your reminder. We have added the coupling constants between ¹⁹F-¹³C in ¹³C NMR.

7. SI: There are some mistakes for MS in SI.

Response: Thanks for your reminder. We have checked and revised the mistakes for MS in SI.

8. Related works on the catalytic prenylation and reverse-prenylation of indoles, such as *Angew. Chem. Int. Ed.* 2019, 58, 5438, are suggested to cite.

Response: Thanks for your suggestions. *Angew. Chem. Int. Ed.* 2019, 58, 5438 has been cited as ref. 26.

Reply to the comments by reviewer 2

1. It would be interesting to see some other radical precursors viz. acids, boronic ester, or trifluoro borates as controls! Further, since the chosen radical precursors of choice were silanes, what about benzyl or alkyl ones?

Response: Thanks for your positive comments and kind suggestions. We investigated some other radical precursors to explore the structural diversity. It was found that the acid derivative (*N*-phenylglycine) underwent

hydroalkylative dearomatization of indole **1d** smoothly, however, the Ireland-Claisen rearrangement was completely suppressed (eq. a). On the basis of a related work by Glorius (*Chem. Sci.* **2021**, *12*, 2816–2822) using boronic esters as radical precursors, we conducted a similar experimental procedure with *N*-Ts indole **1d** and cyclohexylboronic ester, affording the desired dearomative reverse-prenylation product in 32% yield over 3 steps (eq. b). When employing the benzyl trifluoro borate as a radical precursor, the desired dearomatization/rearrangement product was not detected, but an unexpected desulfonation of the *N*-Ts indole happened even in the presence of TMSCl (eq. c). Besides, we also attempted to evaluate the reaction of benzyl silane and *N*-Ts indole **1d**. Under the standard conditions, there was no dearomatization/rearrangement product. Further optimization of the reaction conditions by varying photocatalysts and solvents was still failed to obtain the corresponding product, probably due to the high oxidation potential of benzyl silanes (eq. d).

a) acid as the radical precursor

b) boronic ester as the radical precursor

c) trifluoro borate as the radical precursor

d) benzyl silane as the radical precursor

2. The scope is also well-studied and broad. As indicated in the text, electron-poor indoles are required for the transformation. When considering substitution patterns on indoles, the authors only report two halogen substituted structures (3x and 3y). Are those the only tolerated substitutions? How do more electron-rich groups in those positions behave?

Response: Thanks for your kind suggestions. Reviewer 2 raised the same concern about the substitution patterns on indoles. To demonstrate the substrate scope, more examples for substituted indoles were investigated, including electron-rich and electron poor groups (5-Cl, 5-Br, 5-CN, 5-OMe, 5-Me, 6-OMe, 6-Me and 7-azaindole). The corresponding reverse-prenylated indolines were obtained in good yields with exclusive diastereoselectivity (46–73% yields, >20:1 d.r., **3x–3ae**).

substituted indoles

3x, R'' = Cl, 66%
3y, R'' = Br, 73%
3z, R'' = CN, 64%
3aa, R'' = OMe, 54%
3ab, R'' = Me, 52%

3ac, R* = OMe, 46%
3ad, R* = Me, 58%

3ae, 70%

3. Some additional references on radical indole difunctionalizations are missing (e.g. Nat Commun. 2020, 11, 3263, Org. Lett. 2022, 24, 9386...).

Response: Thanks for your suggestions. We have added some references on radical indole difunctionalizations as ref. 53 – ref. 56.

Reply to the comments by reviewer 3

1. You and co-authors have realized the enantioselective dearomative prenylation of indole derivatives (nature catalysis, 2018, 1, 601), while this work only achieved the diastereoselective dearomative prenylation of indole derivatives.

Response: Thanks for your comments. The You group have reported an enantioselective dearomative prenylation of indole derivatives via a palladium precursor and a chiral phosphoramidite-catalysed asymmetric allylic substitution reactions. It was an elegant methodology that focused on the dearomatization of electron-rich indoles via an ionic $2e^-$ activation mode. Distinct from You's strategy, we show a photoredox-catalysed dearomatization of electron-deficient indoles via a radical $1e^-$ process. Both reaction mechanism and indole substrate scope, even prenylation and reverse-prenylation, are different from You and co-authors' report. In our view, this work is complementary to the known dearomative prenylation of electron-rich indoles. Therefore, it is not an overview assessment that emphasizes only diastereoselectivity versus enantioselectivity, while ignoring the objective differences between these two approaches. However, we can understand the reviewer's concern. Actually, the investigation of catalytic asymmetric version of dearomative prenylation of electron-deficient indoles via a visible-light photoredox catalysis is undergoing in our lab.

2. Although the method uses photocatalysis, it is actually more cumbersome than the reported methods. The method is a two-step continuous reaction, but the second step needs to change the reaction conditions, and it need esterification and N-protection steps to readily prepare the radical acceptor.

Response: In most cases, the dearomatization/rearrangement process underwent spontaneously at the visible-light-activated stage, thus the second heating-step was not essential. To keep the conditions' uniformity and completely promote the rearrangement, we conducted the heating step after the first visible-light-irradiated step. Actually, it is not a cumbersome or complex procedure, which only needs to turn off the light and heat at 60 °C for

3 hours or even without heating. As for esterification and N-protection, they are quite normal steps to prepare the substrates using the classical methods without any tedious or harsh conditions. For example, the reported methods via transition metals-catalysed allylic substitution reactions also required the conversion of allylic alcohol to its carbonate as an allyl electrophile reagent.

3. The methods of Carreira, You and Stark have realized the construction of complex fused-heterocycle skeletons and even natural products, while this method is only suitable for the construction of indoline derivatives with simple structure.

Response: Indeed, Carreira, You and Stark have realized the construction of complex fused-heterocycle skeletons and applied the corresponding approaches to natural product synthesis. Although the transition-metal catalysis (Ir, Pd, etc) with a two-electron transfer process is powerful in tuning the reactivity and selectivity, there are still some challenges in the tandem prenylation/cyclization process. For example, the prenylation/intermolecular dearomatization to achieve indole difunctionalizations was not involved in their research.

Herein, we introduced a conceptually distinct photoredox-catalysed radical strategy to prepare various prenylated and reverse-prenylated indoline derivatives. An array of structurally diverse amines including complex modified natural products and pharmaceuticals were employed as radical precursors, and were readily incorporated in indolines with high functional compatibility and isolated yields and excellent diastereoselectivity. We believe that these indoline derivatives are not simple structures, which are inaccessible or difficult to prepare with conventional approaches.

4. The authors emphasizes “Conventional approaches depend on the use of electron-rich indoles, this method achieved (reverse-)prenylation of electron-deficient indoles”, in fact, the reaction process of this method requires the participation of carbonyl group on indoles, its reaction mechanism depends on the carbonyl group. This is also a limitation of the method because it is not applicable to the substrate without carbonyl group. Other methods have different reaction mechanisms, which do not mean that the dearomatization of electron-deficient indoles is difficult to achieve.

Response: In this manuscript, we never claim that the dearomatization of electron-deficient indoles is difficult to achieve. The present research status is that the available dearomatization reaction types of electron-deficient indoles are still limited in comparison with electron-rich indoles. One of the core processes in our strategy is the Ireland-Claisen rearrangement. As a classical named reaction, it provides a convenient approach to convert the carboxylic esters into the α -alkylated carboxylic acids. We have to admit that Ireland-Claisen rearrangement cannot be performed on substrates without a carbonyl group.

5. The antitumor activity of the synthesized indolines are too weak ($IC_{50} > 5 \mu M$) compared with reported kinase

inhibitors ($IC_{50} < 50$ nM) for the human leukemia. The description of “compounds 3k and 3df displayed superior anticancer activity” is not appropriate.

Response: We agree with the reviewer’s assessment that the description of “compounds 3k and 3df displayed superior anticancer activity” is not appropriate. In order to accurately describe the anticancer activity of compounds 3k and 3df, the “superior anticancer activity” was revised as the “potential anticancer activity”.

REVIEWERS' COMMENTS

Reviewer #1 (Remarks to the Author):

In this revision, the authors have explored more examples of substituted indoles and obtained good results. Other corresponding mistakes and comments have also been corrected or addressed. Therefore, it is suitable for publication on Nature Communication.

Reviewer #2 (Remarks to the Author):

All of the reviewers' suggestions and comments have been effectively addressed in the amended version of the MS by Feng, Liu, and colleagues. I just noticed that the findings from the reviewing process were left out of the MS, which is fine, but these results would be helpful to the readers and would be worthy of a spot in the SI of this MS. I would accept this MS for publication in Nature Communication if that is taken care of.

Reply to the comments by reviewer 1

In this revision, the authors have explored more examples of substituted indoles and obtained good results. Other corresponding mistakes and comments have also been corrected or addressed. Therefore, it is suitable for publication on Nature Communication.

Response: Thanks for your positive comments.

Reply to the comments by reviewer 2

All of the reviewers' suggestions and comments have been effectively addressed in the amended version of the MS by Feng, Liu, and colleagues. I just noticed that the findings from the reviewing process were left out of the MS, which is fine, but these results would be helpful to the readers and would be worthy of a spot in the SI of this MS. I would accept this MS for publication in Nature Communication if that is taken care of.

Response: Thanks for your positive comments. Following your kind suggestions, the corresponding experiments on other radical precursors have been added as an individual spot as "part 5.1" in the Supplementary information.